# Cannabinoids in the Older Person: A Literature Review

**DOI:** 10.3390/geriatrics5010002

**Published:** 2020-01-13

**Authors:** William Beedham, Magda Sbai, Isabel Allison, Roisin Coary, David Shipway

**Affiliations:** 1Medical Student, University of Birmingham Medical School, Birmingham B15 2SG, UK; WXB544@student.bham.ac.uk (W.B.); IVA553@student.bham.ac.uk (I.A.); 2Consultant Physician and Perioperative Geriatrician, Guy’s and St Thomas’ NHS Foundation Trust, London SE1 9RT, UK; magda.sbai@gstt.nhs.uk; 3Specialist Registrar in Geriatric and General Medicine, St James’s Hospital, P.O. Box 580 Dublin, Ireland; coaryr@tcd.ie; 4Consultant Physician and Perioperative Geriatrician, North Bristol NHS Trust, BS10 5NB& Honorary Senior Clinical Lecturer, University of Bristol, Bristol BS8 2PL, UK

**Keywords:** cannabinoids, elderly, THC, CBD, effectiveness, safety, literature review

## Abstract

Introduction: Medical cannabinoids have received significant mainstream media attention in recent times due to an evolving political and clinical landscape. Whilst the efficacy of cannabinoids in the treatment of some childhood epilepsy syndromes is increasingly recognized, medical cannabinoids may also have potential clinical roles in the treatment of older adults. Prescribing restrictions for medical cannabinoids in certain jurisdictions (including the UK) has recently been relaxed. However, few geriatricians have the detailed knowledge or awareness of the potential risks or rewards of utilizing cannabinoids in the older person; even fewer geriatricians have direct experience of using these drugs in their own clinical practice. Older persons are more likely to suffer from medical illness representing potential indications for medical cannabinoids (e.g., pain); equally they may be more vulnerable to any adverse effects. Aim: This narrative literature review aims to provide a brief introduction for the geriatrician to the potential indications, evidence-base, contra-indications and side effects of medical cannabinoids in older people. Methods: A search was conducted of CENTRAL, Medline, Embase, CINAHL and psycINFO, Cochrane and Web of Science databases. Reference lists were hand searched. Abstracts and titles were screened, followed by a full text reading of relevant articles. Results: 35 studies were identified as relevant for this narrative review. Conclusions: Cannabinoids demonstrate some efficacy in the treatment of pain and chemotherapy-related nausea; limited data suggest potential benefits in the treatment of spasticity and anxiety. Risks of cannabinoids in older patients appear to be moderate, and their frequency comparable to other analgesic drug classes. However, the quality of research is weak, and few older patients have been enrolled in cannabinoid studies. Dedicated research is needed to determine the efficiency and safety of cannabinoids in older patients.

## 1. Introduction

Medical cannabinoids have received significant attention in recent years, owing to legal changes in the status of the drugs for medicinal use in various European and North American jurisdictions. Despite this, few European clinicians have experience in the prescription of medical cannabinoids; this is partly attributable to a lack of experience around indications and side effect profiles. In this narrative review, we aim to provide a summary of the current evidence pertaining to the use of medical cannabinoids and any potential risks which must be considered in the older person. 

Although the criminalization of cannabis has historically hindered robust scientific research, cannabis for medical use is now legal in 11 European Union (EU) states, Australia, Canada, Israel and 31 of the 52 states within the USA. As a result, evidence of its efficacy and side effects is emerging, and may inform future medicinal use.

To date, the majority of cannabinoid research has focused upon a healthy younger population, free from frailty and major comorbidity. However, advancing age is unequivocally associated with the accumulation of medical comorbidity and individual disease states, such as cancer. It is therefore clear that older persons may become an important group of potential cannabinoids users in the future. Greater awareness amongst geriatricians of the potential indications and hazards of cannabinoids in the older patient is therefore imperative. 

## 2. Background

The medicinal properties of cannabis have been recognized since at least 1500 BC, when the drug was listed in the ancient Chinese pharmacopeia as a treatment for rheumatic pain [1]. In the mid-nineteenth century, cannabis smoking was introduced into western medicine as a treatment for rheumatism, before becoming prohibited in the UK in 1925 for general use and in 1971 for medicinal use.

Plants of the cannabis family contain two major cannabinoids: tetrahydrocannabinol (THC) and cannabidiol (CBD). THC is the primary psychoactive compound, and appears to have neuroprotective, anti-inflammatory, anti-emetic and analgesic effects. CBD has no psychoactive properties, and appears to attenuate the psychoactive properties of THC when used in combination. CBD in isolation reduces proinflammatory cytokine release and displays antioxidant properties [2].

Two strains of the cannabis plant (*sativa* and *indica*) are considered to be of potential medical use. Each strain has different properties, with the THC:CBD ratio determining their clinical effects. This ratio is used to classify cannabis plants according to chemical constitution, with 3 *chemotypes* currently recognized [3]. Chemotype-1 plants are those which contain the highest concentration of THC, and are used recreationally due to their potency and psychotropic effects. 

In humans, cannabinoids act via the modulation of the endocannabinoid system, which consists of CB1 and CB2 cannabinoid receptors. The expression of CB1 and CB2 receptors varies between tissues; CB1 receptors are primarily located within the central nervous system (CNS); CB2 receptors are abundant in the peripheral tissues and appear to have a greater effect upon cytokine action and modulation of the immune system [4]. THC and CBD differ in their affinity to CB1 and CB2 receptor agonists; THC has an affinity for both receptors, whereas CBD has more selective affinity for CB2 receptors. The physiological function of the endocannabinoid system remains incompletely understood. However, current research suggests that this system is involved in: neuroprotection, nociception, autonomic tone, immune function, connective tissue repair, satiety and behavior [5].

## 3. Aims

It is important to recognize that the heterogeneity and quality of the research methodology in this field makes interpretation of research in this area problematic. Firstly, it should be noted that analysis of the efficacy or safety of smoke inhalation is methodologically challenging, due to the uncertain composition of smoked cannabis. Importantly, illicit recreational cannabis strains have been selectively bred to increase THC concentrations (“*skunk*”) resulting in augmented neuropsychiatric side effect profiles. 

These recreational agents receive considerable attention in the mainstream media, but their toxicity should not necessarily be assumed to be transferrable to the standardized production of medicinal cannabinoids, and their use in a medical context in a controlled jurisdiction [6]. Secondly, the heterogeneity of cannabinoid products, the mode of administration, and disparate population group makes a systematic review in this context of limited value; it is highly unlikely that firm conclusions would be possible from the quality and quantity of evidence currently available in the older adult. This narrative literature review therefore aims to provide a concise introduction and summary for the geriatrician. It aims to provide an overview of potential indications, contra-indications and probable side effects of medical cannabinoids in older people. 

## 4. Methods

In order to minimize the bias in our narrative review, a literature search was conducted in April 2019. Bibliographic databases that were searched included: CENTRAL, Medline, Embase, CINAHL and PsycINFO. Further database searches included: the Cochrane Library and Web of Science. Relevant, unpublished literature was also sought through clinicaltrials.gov and the International Clinical Trials Registry Platform (ICTRP). Finally, for completeness, the reference lists of relevant studies and systematic reviews were hand-searched for additional references.

All databases were searched using a unique combination of MeSH terms and title key-word searches. MeSH terms included: “Cannabinoid OR Cannabis OR Medical marijuana” AND “Aged or Geriatrics”. Keywords were then added, where appropriate, to improve the scope of the search, and included: “CBD OR THC OR Epidiolex OR Nabilone OR Nabiximol* OR Sativex” AND “Old OR Elderly OR Age* OR Geriatric*”. These keywords and MeSH terms were combined with the appropriate Boolean operator within each database to form a comprehensive search strategy.

Where possible, in each database, additional search restrictions were applied to ensure an adequate focus, which included: human studies, elderly population, publication in the last 20 years and publication in the English language. 

## 5. Results

The database search yielded 581 results. A further 22 records were identified through other sources. Title and abstract screening were conducted by two independent researchers. Any discrepancies of the inclusion decision were resolved through discussion. Following the screening process, 57 articles remained. All were read in full by the primary author. Following full text reading, 38 were deemed relevant for this review. The main findings of these studies can be found in Table 1. This selection process is illustrated below in Figure 1.

## 6. Potential Indications for Medicinal Cannabinoids in the Older Person 

In addition to smoked cannabis, there are currently three cannabinoid preparations which are in widespread use, namely, nabilone, dronabinol and nabiximols (Sativex). 

## 7. Pain

### 7.1. Smoked Cannabis

Although smoked cannabis is widely known to be used by individuals as self-administered analgesia, this route of administration is difficult to study robustly; limited data are therefore available to support this mode of drug delivery. To address this, a 2010 randomized controlled trial was conducted to assess the efficacy of smoked cannabis within a population of patients suffering from chronic neuropathic pain. This study found a dose–response relationship between the potency of inhaled cannabis and pain amelioration, demonstrating a statistically significant improvement in average daily pain scores between the highest potency and control groups [46]. 

### 7.2. Nabilone

Nabilone is a synthetic analogue of THC. It is currently endorsed in the UK by NICE for chemotherapy-induced nausea and vomiting refractory to other treatment [47]. However, it has also been reported to display analgesic properties. In 2017, the Canadian Agency for Drug and Technologies in Health (CADTH) published a critical appraisal for Nabilone use in chronic pain [48]. CADTH concluded that, although there are limitations in the current evidence base, Nabilone can improve pain, with limited associated harms. 

This conclusion was largely based upon the results of a 2015 systematic review, which examined four studies evaluating nabilone for chronic pain in diabetic neuropathy, multiple sclerosis, fibromyalgia and a medication-overuse headache. In all conditions except fibromyalgia, Nabilone was found to be superior to control for improving pain. CADTH also cited a further study, relevant to this review, in which the average participant was aged 64 years. However, this study found that in post-radiation neuritis, there was no significant improvement observed.

### 7.3. Dronabinol

Dronabinol is a synthetic THC analogue that has been approved for pain control in some jurisdictions. Narang et al. reported the effect of dronabinol on pain in a placebo-controlled trial [39]. In this study, dronabinol was used as an adjuvant to opioids with a significant reduction in pain at 8 h noted in the dronabinol arm. These findings continued into the second open-label phase of the trial, where a statistically significant reduction in the pain score was observed from baseline in the dronabinol arm. However, other trials have not been able to reproduce these results over a longer follow up period [22]. 

### 7.4. Nabiximols

Unlike the THC analogue nabilone and dronabinol, Nabiximols is a whole plant extract of cannabis, containing both THC and CBD. It is currently licensed for the treatment of multiple sclerosis (MS)-associated spasticity and as an adjuvant for cancer pain treated with opioids.

In both Israel and Canada, nabiximols has also been licensed for neuropathic pain in multiple sclerosis. CADTH has also published a critical appraisal of THC–CBD combination buccal sprays used for chronic neuropathic or non-cancerous pain [49]. The five systematic reviews cited by CADTH provided conflicting evidence: some reported significant improvement in pain, others no improvement. CADTH reached the conclusion that short-term benefits can be seen with the use of nabiximols for patients with chronic-neuropathic or non-cancer pain, but evidence for its continued use is unclear.

Further recent evidence evaluating the use of cannabinoids for treating non-cancer pain can be derived from a 2018 systematic review. This included 104 randomized and non-randomized trials reporting on cannabinoid use for pain. A total of 45 RCTs assessed the impact of cannabinoids on altering pain intensity. 30 studies were included in a meta-analysis, which found that cannabinoid treatment resulted in a significantly greater pain reduction than controls [17].

A recent Cochrane review also found that cannabis-based products were more effective than the placebo at reducing pain severity. However, the lack of high-quality evidence was noted, guarding the grade of recommendations [14]. It is important to appreciate that no studies have specifically aimed to evaluate the effect of these drugs on pain in older people. Extrapolation of adult data to older adults is therefore largely speculative, and dedicated research in this group is indicated.

## 8. Chemotherapy Related Nausea

The use of cannabinoids to manage chemotherapy-related nausea is relatively well researched. A recent Cochrane systematic review and meta-analysis found that cannabinoids were effective and performed better than the placebo [27]. Cannabinoids have also been shown to display equal efficacy when compared to conventional antiemetic drugs, albeit with an increased side effect profile including dizziness, sedation and dysphoria [25]. Notwithstanding the side effect profile, cannabinoids may have a role to play in refractory chemotherapy-induced nausea. 

## 9. Spasticity

Spasticity is a common consequence of stroke and other primary neurological disorders. Epidemiological evidence indicates that 20% of patients suffering from Multiple Sclerosis (MS) have used cannabis to ease their symptoms of stiffness [50]. 

A 2018 systematic review in MS patients found that cannabinoids reduced self-reported spasticity compared to the control; these self-reported findings were not however reflected in the objective spasticity scores [15]. It remains unclear whether cannabinoids have a role to play in the management of spasticity due to MS, or other neurological injury. 

## 10. Appetite Stimulation

The evidence for appetite stimulation in the context of cancer-induced anorexia is currently limited. A 2002 trial compared the cannabis extract dronabinol to megestrol (synthetic progestin). This study found that dronabinol was less effective than megestrol at appetite stimulation and inducing weight gain [42]. Similarly, a 2006 trial by the Cannabis in Cachexia Study group was terminated due to insufficient differences between the study arms of: (1) cannabis extract, (2) tetrahydrocannabinol and (3) placebo [40]. Currently therefore, there is no evidence to indicate that cannabinoids have any effect upon cancer-induced anorexia.

Anorexia in the elderly is complex, and the mechanisms underlying the anorexia of ageing may be distinct to the anorexia of dementia or cancer. Few trials have evaluated the efficacy of cannabis products for anorexia associated with ageing or dementia. A small placebo-controlled cross-over trial found measurable changes in weight gain and decreased agitation with dronabinol [45]. However, this study was limited by small numbers of patients at its follow-up, and at present the European Society for Clinical Nutrition and Metabolism has concluded that there is currently no evidence to support the use of cannabinoids in the cachexia of dementia [51].

Further work is required to determine whether cannabinoids have any role in stimulating appetite and weight gain in patients with anorexia related to underlying dementia.

## 11. Behavioral Problems in Dementia

The neuropsychiatric symptoms of dementia have been evaluated in one double-blind placebo-controlled study. This found that although THC administration was well tolerated, it had no effect on neuropsychiatric symptoms when compared to the placebo. However, the authors did speculate that due to the tolerability of side effects, further work was required to assess the efficacy of higher doses [9]. 

Two systematic reviews have also been published in this field [23,28]. When pooled, these reviews included five studies which assessed dronabinol; 1 assessing nabilone; 2 THC. All studies assessing dronabinol and nabilone reported improvements in behavioral symptoms; neither studies assessing THC demonstrated improvements. Dronabinol and nabilone may therefore have a potential role in dementia-related behavioral problems. 

The safety and tolerability of cannabinoids in dementia has also been evaluated further [26]. One study evaluated the safety of oral THC through monitoring adverse events and vital signs over 12 weeks. During follow up, there were fewer adverse events within the THC group compared to placebo control. Furthermore, only six cases of adverse events were attributed to THC use, all of which were considered to be mild in nature. These adverse events included: agitation, dizziness and fatigue. From these data, the authors concluded that cannabinoid use has a tolerable safety profile in older people with dementia, with doses of up to 5 mg/day being less likely to induce adverse effects. 

## 12. Anxiety

Anxiety disorders increase with age and are associated with increased healthcare needs and social isolation [52].

Patients taking recreational cannabis report conflicting findings in relation to anxiety. This may be characterized by a curtailment of their anxiety, but also potential augmentation and paranoid thought. This paradoxical observation is thought to be a reflection of the differing CBD:THC ratios present in the smoked plant extract; in general CBD is thought to reduce anxiety, with THC having an antagonistic effect [1].

A 2009 randomized, double-blind, placebo-controlled study found that CBD administration was associated with diminished autonomic arousal on fMRI and a reduced anxiety state; conversely a state of increased autonomic arousal and increased anxiety was found with THC administration [38]. A further randomized control trial conducted in seasonal affective disorder patients found that the administration of CBD prior to the introduction of anxious stimuli significantly reduced anxiety [36]. CBD may have potential for diminishing anxiety in both pathological and non-pathological anxiety states.

## 13. Movement Disorders

A 2014 open label study found that 30 min after smoking cannabis, patients reported improvements in the major symptoms of Parkinson’s Disease (PD), including: rigidity, tremor, pain, sleep and bradykinesia [31]. This study design displays many methodological problems which render the results challenging to interpret. However, further exploration of the use of smoked cannabis for the treatment of PD is warranted. 

A 2017 review consisting of four randomized, double-blind, placebo-controlled trials further evaluated the effect of cannabinoids in PD [24]. Two trials detected positive outcomes for the use of cannabinoids in PD; two did not. One study detected a significant reduction in levodopa-induced dyskinesia and rush dyskinesia [43]. Significant improvements in the Parkinson’s disease questionnaire (PDQ-39) and activities of daily living scores were also reported [32]. Subsequent studies have not been able to replicate these findings. Further work is required to determine whether cannabinoids affect the symptoms of movement disorders.

## 14. Potential Hazards of Medical Cannabinoids in Older Adults

### 14.1. Smoked Cannabis 

We are unaware of any RCTs that assess the safety of smoked cannabis in an elderly population. One observational study has found that cannabis smoking in elderly individuals was associated with emergency department attendance and injury [53]. More generally, smoked cannabis had been linked to an increased risk of psychosis and neurocognitive dysfunction in younger adults and adolescents, though the relevance of these observations in older adults is currently unknown [54]. In the context of reduced neurocognitive reserve, it seems plausible that older adults are likely to be at increased risk of neurocognitive side effects. 

### 14.2. Nabilone 

Nabilone’s safety is also poorly documented in an elderly population. A case series observed the side effects of using nabilone in elderly patients with neuropsychiatric symptoms. This study observed sedation as the primary side effect in this population [55].

### 14.3. Dronabinol 

The safety of Dronabinol in an elderly population has only been observed in two studies [45,54]. These studies found that the most common side effects were sedation, anxiety, emotional labiality, fatigue, somnolence and euphoria. A serious side effect of a single patient experiencing a grand mal seizure was also observed [56].

### 14.4. Nabiximols

Currently no studies specifically address the safety of nabiximols in an elderly population. However, adult data indicate more adverse events in the nabiximols group compared to placebo [57]. The most common adverse events reported were: asthenia, vertigo and nausea, and it seems plausible that these side effects may be applicable to older patients

## 15. Safety and Adverse Effects

Despite numerous small studies describing a modest side effect profile of cannabinoids, we are aware of only one systematic review evaluating the safety profile of cannabis and cannabis products in the elderly [29]. This systematic review found that cannabinoids were associated with a higher rate of adverse events than the control. 

The most common side effects experienced by users of cannabinoids included drowsiness, dry mouth, coordination disturbance and headache [30]. However, no severe adverse events were reported in this review, and only 1.4% of participants discontinued treatment due to adverse effects.

A more recent prospective study addressed the safety and efficacy of cannabis specifically in older subjects [11]. This study reported that in the 6-month follow up period, 31.7% of the cohort experienced at least one side effect. The most common side effect experienced was dizziness (9.7%). Others included confusion or disorientation (1.9%) and hallucinations (0.8%). The rate of falls amongst participants receiving cannabinoids was not elevated despite these side effects: 53.4% of the 515 participants reported at least one fall in the 6 months before cannabis treatment, with only 21.9% reporting a fall in the 6 months after initiating treatment. However, there are no data reporting the respective frailty of individuals enrolled in this trial, and the clinical impact of increased dizziness in persons vulnerable to falls is currently unknown. 

Notably, the participants of this study smoked cannabis plants with heterogenous THC: CBD formulations. Although there was a predominance of THC-dominant strains used, different formulations reduce the reliability and application of this data to a wider population. 

## 16. Medication Interaction

Cannabinoids including THC and CBD are both metabolized through different isoforms of the P450 enzyme system. Cannabinoids therefore may compete with other drugs which are metabolized through this pathway [16]. Notably, five of the seven P450 isoforms that metabolize CBD also metabolize warfarin. Cannabinoid use may therefore increase serum warfarin concentration and bleeding risk. Despite an in vitro study confirming this interaction, there is currently only one case report citing this; it is therefore currently unclear what the clinical significance of drug interaction may be in recreational users. However, in the context of regular medication for chronic illness, this may become of more clinical importance [35]. Other drug classes that may potentially be influenced by competition for cytochrome P450 metabolism are listed in Table 2. 

## 17. Long Term Safety

The World Health Organization (WHO) recognizes that recreational cannabis use is associated with significant cardiovascular risks. These effects are less well explored in the context of medicinal cannabis. A 2017 systematic review found that in individuals with pre-existing cardiovascular disease, the greatest risks posed by cannabis products were ischemic strokes or myocardial infarctions [34], though wide scale data evaluating this area is currently absent. It has been suggested that cannabinoids may be associated with an increase in the risk of solid tumor neoplasia. At present to our knowledge, no definite data exist to confirm this.

Although it is clear that the side effect profile of cannabinoids is inadequately understood at present, it is likely to be significant, especially in older people who have to date been excluded from clinical trials in this field. Notwithstanding this current clinical uncertainty, it is important to reflect that the side effect profiles of other drug classes necessarily used in older adults (e.g., opiates, NSAIDS, anticonvulsants) are equally problematic. Limited data indicate that cannabinoids may potentially reduce the opiate requirements in some pain settings, or facilitate withdrawal from opiates in their entirety [11]. Further dedicated research in the frail, older patient is urgently warranted to quantify the risks and potential benefits. 

## 18. Conclusions

Cannabinoids appear to demonstrate some moderate efficacy in the treatment of pain and chemotherapy-related nausea; there are limited data to suggest any possible benefits in the treatment of spasticity and anxiety. However, the quality of research in this area is weak and derived from heterogeneous interventions and study populations, making interpretation of the sparse literature in this field difficult. 

The existing evidence base appears to indicate that the risks of cannabinoids in older persons are modest, but not insignificant. High quality research is urgently needed to further determine the efficacy and side effect profile of cannabinoids in older persons and compare them to current alternatives.

## Figures and Tables

**Figure 1 geriatrics-05-00002-f001:**
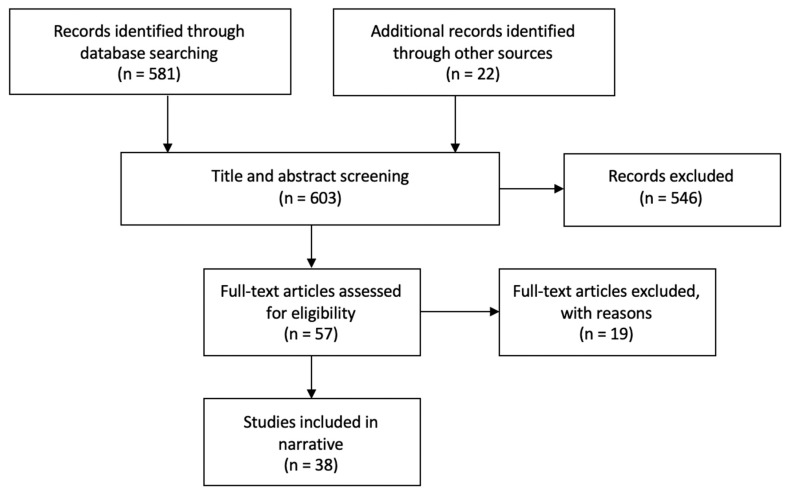
Study selection flow diagram.

**Table 1 geriatrics-05-00002-t001:** Search results main findings.

Author	Year	Study Type	Main Findings
Minerbi A. [7]	2019	Review	Evidence for efficacy is sparse, and considerable adverse effects have been documented on cognitive, cardiovascular and gait function.
McMaster University [8]	2019	RCT protocol	Protocol for a trial that will assess the efficacy of CBD versus a placebo for post-operative pain following knee arthroplasty.
Van Den Elsen G. [9]	2019	RCT	THC had no significant effect on neuropsychiatric symptoms, but was well tolerated by participants.
Beauchet O. [10]	2018	Mini review	Some studies report improved outcomes. However, studies are not focusing on an elderly population, and are of limited quality, making any conclusions difficult to draw.
Abuhasira R. [11]	2018	Prospective study	Therapeutic cannabis is both effective and safe in an elderly population. In particular, its ability to reduce other prescriptions, such as opioids, is advantageous.
Agornyo P. [12]	2018	Survey	Regular cannabis use in older patients resulted in reduced pain, reduced medication side effects and the discontinuation of other pain medications.
Briscoe [13]	2018	Review	The volume of current research is increasing, but there is still insufficient evidence to make robust recommendations.
Mücke M. [14]	2018	Systematic review	The risks of cannabinoid treatments for neuropathic pain appear to outweigh any potential benefits.
Torres-Moreno M. [15]	2018	Systematic Review	Results favored cannabinoids over the placebo. However, there is little evidence to suggest that cannabinoids are effective for bladder dysfunction, pain and spasticity in MS patients.
Grayson L. [16]	2018	Case report	Observed reaction between cannabidiol and warfarin. INR more than doubled.
Stockings E. [17]	2018	Systematic Review	The effectiveness of cannabinoids is minimal for non-cancer pain. Studies show very high NNT for cannabinoids.
Mahvan T.D. [18]	2017	Review	There are considerable risks for elderly patients using cannabis, especially those with comorbidities.
Katz I. [19]	2017	Review	For Parkinson’s Disease (PD), cannabis can be useful as a last resort. For dementia, cannabis can help with behavioral problems. Sleep disturbance and weight loss are also areas of potential benefit from cannabis.
Bellnier T. [20]	2017	Retrospective study	Cannabis led to improved quality of life, reduced pain, reduced opioid use and led to reduced costs.
Hansra D.M. [21]	2017	Survey	Dronabinol led to significant improvements in QoL, appetite and anxiety, amongst cancer patients.
Schimrigk S. [22]	2017	RCT	Dronabinol was non-significantly better at reducing pain in MS patients. However, Dronabinol was associated with much higher rates of adverse events.
Lim K. [23]	2017	Systematic Review	Studies indicate that cannabinoids could potentially benefit anxiety, psychotic symptoms, PTSD and dyskinesia in PD.
Stampanoni M. [24]	2017	Review	The evidence for cannabinoids in PD patients is currently inconclusive.
Tafelski S. [25]	2016	Systematic review of systematic reviews	Cannabinoids should not be recommended for chemotherapy-induced nausea and vomiting, as either a first- or second-line treatment. Anti-emetics are currently superior.
Ahmed A.I. [26]	2015	RCT	THC was well tolerated by elderly participants with dementia. There were more adverse events within the placebo group. Adverse events experienced by THC users were most commonly: dizziness, fatigue and agitation.
Smith L.A. [27]	2015	Systematic review	Cannabinoids may have potential at reducing chemotherapy-induced nausea and vomiting.
Liu C. [28]	2015	Review	Some studies show significant improvement in agitation and aggression in Alzheimer’s Disease (AD) patients, but evidence is insufficient to make robust conclusions.
van den Elsen G.A. [29]	2014	Systematic Review	Studies show that THC may be useful for anorexia and behavioral symptoms in dementia. Adverse symptoms were common in the cannabinoid groups, with sedation being the most frequent.
Ahmed A.I. [30]	2014	RCT	THC was associated with more adverse events than placebo. Increasing the dose of THC increased the frequency of adverse events.
Lotan I. [31]	2014	Open label study	Smoking cannabis significantly improved sleep and reduced pain in PD patients.
Chagas M.H. [32]	2014	RCT	Cannabidiol had no significant impact upon PD symptoms, but showed an improved quality of life (QoL).
Ahmed A.I. [33]	2014	Letter	There is scanty evidence for the safety and effectiveness of cannabis and cannabinoids in the elderly. We also cannot extrapolate results from the studies of younger individuals onto an elderly population.
Wolff V. [34]	2013	Case series	It is highly likely that there is an association between cannabis and stroke.
Yamaori S. [35]	2012	In vitro study	THC and CBD both inhibited CYP2C9 activity.
Bergamaschi M. [36]	2011	RCT	CBD significantly reduced anxiety in participants giving public speeches.
Patel T. [37]	2010	Review	Cannabinoid agonists have been shown to reduce chronic pruritis in patients with atopic dermatitis, lichen simplex, prurigo nodularis, and CKD-related itching.
Fusar-Poli P. [38]	2009	RCT	CBD reduces autonomic responses to fear, whereas THC augments the autonomic response.
Narang S. [39]	2008	RCT	Dronabinol resulted in additional pain relief for individuals already using opioids for noncancer-related chronic pain.
Strasser F. [40]	2006	RCT	THC and cannabis extract were no better than the placebo at increasing appetite or QoL for patients with cancer-related cachexia.
Sutton L.M. [41]	2003	Review	Cannabinoids can be useful for dyspnoea at the end of life. However, there must be caution with these agents due to the psychotropic effect and reflex tachycardia and hypotension.
Jatoi A. [42]	2002	RCT	Megestrol acetate performed better than dronabinol for palliating anorexia in cancer patients.
Sieradzan F. [43]	2001	RCT	Nabilone significantly reduced dyskinesias in PD patients.
Yeh S. [44]	1999	Review	Anecdotal evidence has shown Dronabinol to be effective at increasing appetite in HIV patients and Alzheimer’s patients.
Volicer L. [45]	1997	RCT	Body weight increased more within the dronabinol group than with the placebo.

**Table 2 geriatrics-05-00002-t002:** Common CYP2C9 substrates.

Anti-inflammatory drugs e.g., ibuprofen Anti-convulsant drugs e.g., phenytoin Asthma drugs e.g., zafirlukast Anti-cancer drugs e.g., cyclophosphamide Hypoglycemic drugs e.g., glimepiride Anticoagulants e.g., warfarin ACE inhibitors e.g., losartan

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
