# Peer review of "Cannabinoids in the Older Person: A Literature Review"

_geriatrics, 2020, doi:10.3390/geriatrics5010002_

Round 1

Reviewer 1 Report

The aim of the present narrative literature review was provide a brief summary for the geriatrician of the current indications, evidence-base, contraindications and side effects of medical cannabinoids in older people.

Overall, I found the paper timely, very interesting, useful and scientifically sound. However, I have some concerns on it that should be addressed prior publication and these are explained below:

1) I would suggest the Authors add a brief note at the end of Introduction on the reasons why this topic should be an interesting object of review. Even if it may be obvious, I believe that a more detailed explanation of aims and scopes of the Review would be very useful to the reader.

2) The main limitation of this review was the narrative type: why the Authors didn't conducted a systematic review should be explained.

3) The paragraph on "Safety and Adverse Effects" is too short. I believe that every examined drug may be accompanied by a critical evaluation of the safety concerns.

Author Response

Dear Sirs,

Many thanks for your helpful criticism and constructive comments. We have endeavoured to address these and respond forthwith, below.

We very much hope these will render the article fit for publication, but if further improvements can be made, please do not hesitate to let us know.

With kind regards,

David Shipway and Colleagues.

1) I would suggest the Authors add a brief note at the end of Introduction on the reasons why this topic should be an interesting object of review. Even if it may be obvious, I believe that a more detailed explanation of aims and scopes of the Review would be very useful to the reader.

Thank you for the helpful comments, this has been done.

2) The main limitation of this review was the narrative type: why the Authors didn't conducted a systematic review should be explained.

This narrative review aims to provide a basic introduction to an unfamiliar topic to the general geriatrician readership of the journal. The reason we have not performed a systematic review is now laid out in the article in more detail. Ultimately, the heterogeneity of studies, drugs, agents, modes of administration, populations studied and scarcity of data complicate the execution of a systematic review. This means that in our opinion, a narrative introduction to the topic is more useful to the journal readership than a systematic review at the present time, which we consider would be unable to reach any meaningful conclusions. We have included our search strategy and flow diagram- if the editors feel this is helpful- to indicate the lengths we went to minimise the potential bias of the narrative review process.

This has been made clear in the text.

3) The paragraph on "Safety and Adverse Effects" is too short. I believe that every examined drug may be accompanied by a critical evaluation of the safety concerns.

Thank you for your comments. We have expanded our section on safety and adverse events. We have focused mostly on class actions with only specific notable drugs mentioned as this is more in keeping with the scope of the work. This has been expanded as requested to include a breakdown of evidence for each cannabinoid mode of delivery/preparation.

Reviewer 2 Report

Abstract

Please rearrange the contents as follows: a few sentences of introduction, aim, the Method used to Reach the study aim, the main findings and implications of the findings.

Introduction

Please provide a description of canabinoids and their branches etc.

A brief description of the history of studies on this topic is needed.

The end section of paragraph should be the aim.

Methods

What sort of review you have done? Is this systematic?Scoping? enything else?? Clarify this and use appropriately provide the details regarding how the review has been done. 

Many details of the review process are missing. To ensure that the review process is reliable and no article is missed, you must provide all details regarding the review process.

The results section should be devided from the Methods. Please provide a dessription, also, of based on what Logic you have presented the results.

A table would fine to summarise the results of your review.

There must be a discussion section on how Your findings can be Connected to the high Level perspectives on the study phenomenon. The use of policy statements etc for comparison with yours would be fine.

Under conclusion, limitaions, suggestions for future Research and policy making in practice are needed. 

Author Response

Dear Sirs,

Many thanks for your helpful criticism and constructive comments. We have endeavoured to address these and respond forthwith, below.

We very much hope these will render the article fit for publication, but if further improvements can be made, please do not hesitate to let us know.

With kind regards,

David Shipway and Colleagues.

Abstract

Please rearrange the contents as follows: a few sentences of introduction, aim, the Method used to Reach the study aim, the main findings and implications of the findings.

This has been done. Please see attached file.

Introduction

Please provide a description of canabinoids and their branches etc.

This is included in the "Background" paragraph, commencing at line 60. We have limited this detail, because, although interesting, we are mindful of the need for brevity in an article of this nature.

A brief description of the history of studies on this topic is needed.

Again, this is included in the background paragraph, commencing at line 60. We have not elaborated specifically about the history in the UK, because we are aware each European and American jurisdiction has its own history, and it is beyond the scope of this article (and clinical importance) to discuss each history in detail

The end section of paragraph should be the aim.

This has been done, as requested.

Methods

What sort of review you have done? Is this systematic?Scoping? enything else?? Clarify this and use appropriately provide the details regarding how the review has been done. 

This narrative review aims to provide a basic introduction to an unfamiliar topic to the general geriatrician readership of the journal. The reason we have not performed a systematic review is now laid out in the article in more detail. Ultimately, the heterogeneity of studies, drugs, agents, modes of administration, populations studied and scarcity of data, mean that in our opinion, a narrative introduction to the topic is more useful to the journal readership than a systematic review at the present time. We have included our search strategy and flow diagram- if the editors feel this is helpful- to indicate the lengths we went to minimise the potential bias of the narrative review process.

Many details of the review process are missing. To ensure that the review process is reliable and no article is missed, you must provide all details regarding the review process.

In the methods section we have now provided a more detailed explanation of the search terms and methods used to search for articles in this narrative review. However, we would again stress that this article does not purport to be a systematic review, for the reasons we have already explained. In our experience of writing narrative reviews, including those published in this journal, the methods utilised for this narrative review are typical, although admittedly not exhaustive. We acknowledge that it is possible that some articles may have been excluded; however the small volume of relevant publications in this field and low quality is precisely the reason why we conducted a narrative, rather than systematic review. As we have explained, the aim of this article is to provide a basic introduction to the general geriatric reader.

oThe results section should be devided from the Methods. Please provide a dessription, also, of based on what Logic you have presented the results.

This has been done.

A table would fine to summarise the results of your review.

This has been done.

There must be a discussion section on how Your findings can be Connected to the high Level perspectives on the study phenomenon. The use of policy statements etc for comparison with yours would be fine.

In the article we have referenced the statements of respected bodies and their position. We do not presume to offer more definitive recommendations based on the limited evidence available at the current time.

Under conclusion, limitaions, suggestions for future Research and policy making in practice are needed. 

We have made recommendations for further research as suggested, but our intention with the article was never to lobby for changes to government or professional policy or to propose practice guidance. As we have stated, our  intention has been to update the generalist reader in our specialty with an accessible, short review.

Round 2

Reviewer 2 Report

Nothing more.